# Forms of Damage of Bricks Subjected to Cyclic Freezing and Thawing in Actual Conditions

**DOI:** 10.3390/ma12071165

**Published:** 2019-04-10

**Authors:** Teresa Stryszewska, Stanisław Kańka

**Affiliations:** Faculty of Civil Engineering, Cracow University of Technology, Warszawska 24, 31-155 Cracow, Poland; skanka@pk.edu.pl

**Keywords:** ceramic brick, freezing-thawing, destruction form, powdering, flacking, cracking

## Abstract

The paper presents the characteristics of damage of bricks in masonry structures of significant historical value as a result of cyclic freezing and thawing. Based on extensive investigation, which included macroscopic description, determination of compressive strength and tensile strength, determination of the mineral composition, scanning microscopy observation and determination of the porosity structure of bricks, three forms of frost damage were distinguished, termed as powdering, flaking and cracking. Bricks were collected from existing historical buildings 70 years after their construction. It was observed that the particular form of frost damage of bricks is highly correlated with the structure of porosity. Additional factors affecting the form of frost destruction are the strength of the material, its mineral composition and the spatial arrangement of the texture elements. Taking the above into account, it is possible to evaluate frost resistance of bricks and specify the form of damage. Predicting frost resistance and forms of damage based on low-destructive methods using small samples is the expected solution in the case of heritage facilities.

## 1. Introduction

One of the reasons for the deterioration of bricks that are not protected against water action, especially in historical buildings, is the freezing of water within the pores of the material. For bricks, the process of water freezing begins in the largest pores and as the temperature drops it affects pores with smaller diameters. In very fine pores, i.e., with a diameter smaller than 0.1 µm, water freezes in temperatures significantly below 0 °C. The freezing of water causes internal stresses that are the direct cause of the damage. The occurring pressure rises as the temperature falls. It is obvious that when the tensile strength of a brick is lower than the pressure of the crystallising water, cracks appear in the material. Thus, the material factors that determine frost resistance of a material include mechanical properties, particularly its tensile strength and Young’s modulus [1,2]. These parameters define the resistance of a material to ice expansion. Higher strength enhances resistance to stresses which are generated in the water crystallisation process. Of significance is also the mineral composition of the raw materials used in brick manufacturing as well as the phase composition of burnt bricks [3,4]. Products containing clay minerals of the kaolinite and illite group in their original composition are characterised by better frost resistance than materials with prevailing content of clay minerals of the montmorillonite group in their raw material composition [5]. According to [6], brick resistance is closely related to the degree of vitrification of the texture. However, due to the difficulty in assessing the content of this phase, this feature should only be treated indicatively and in connection with other parameters, such as strength and porosity.

The other factors that determine frost resistance include physical properties, such as permeability and porosity. Permeability enables and determines free water flow through a porous material without destroying the texture.

Most researchers, however, consider frost resistance of bricks conditional on its porosity and particularly on the structure of porosity [1,7,8,9,10,11]. Furthermore, investigation results and mathematical models presented by [12] suggest (taking account of the analogy between pressure from salt crystallisation and pressure from freezing water) that frost resistance also depends on the shape of the pores. Despite many studies, the impact of the structure of porosity on frost resistance is still not unequivocal and the quest for some tangible correlations still remains the area of interest of many researchers. According to [13], lack of frost resistance is particularly related to the presence of smaller pores, i.e., with diameters below 1.4 µm. Bricks with a content of minimum 40% of pores with diameters larger than 0.25 µm or with a content of approx. 40% of pores with diameters larger than 1.4 µm are characterised by good frost resistance [13]. If the prevailing pores in the material are in the range of 0.25–1.4 µm, the frost resistance of the material is low. Robinson [14], in turn, observed that materials with predominant fraction of pores with diameters below 1 µm have low frost durability in contrast to materials in which larger pores with diameters over 2 µm are prevalent. In [1,15], it has been demonstrated that pores with diameters greater than 3 µm have essential influence on improving frost resistance. Some argue that even with great porosity but with prevalent presence of pores over 3 µm bricks are characterised by good resistance to frost [6,15,16,17,18]. This is due to the fact that large pores form a kind of buffer against the stresses that accompany ice crystallisation [16]. Furthermore, crystallisation pressure in larger pores is lower than in the smaller pores [18]. According to [19], in turn, the most dangerous pores are characterised by diameters less than 0.5 µm and their percentage in pores with diameters up to 0.74 µm is an essential factor determining frost resistance. The destructive effect of pores with diameters below 1 µm was confirmed in studies by [20]. The authors demonstrated that in historical buildings, with a very long exposure time, an essential factor is the fraction of the smallest pores which initially do not compromise frost resistance of the bricks but are precursors of larger pores. They have shown that the structure of porosity of bricks does not change over the years. As an effect, small pores with diameters below 1 µm (the presence of which is not harmful with respect to frost resistance) increase their diameters and thus cause the structure of porosity to move towards larger pores with diameters in the range 1–5 µm, which compromise frost resistance. In the light of these studies, pores with diameters less than 1 µm increase the susceptibility of the material to frost damage.

Discussions on frost resistance tend to leave out the aspect concerning the form of brick damage which is the visual effect of the process of frost damage. In historical buildings with long-term exposure to climatic conditions characterised by varying temperatures, with a great number of fluctuations around freezing point, brick damage is a natural, very frequently observed phenomenon. It assumes various forms. Based on the authors’ own observations, three forms of frost damage were distinguished; i.e., powdering, flaking and cracking. The objective of the investigation presented in the paper and of the analysis of its results is to determine the correlation between a particular form of frost damage of bricks and their properties.

## 2. Scope of Research 

### 2.1. Materials Tested

The tests and observations presented in the paper were conducted for materials collected from masonry structures of the former extermination camp Auschwitz II–Birkenau in Oświęcim [21]. That camp was set up during the Second World War by German Nazis. It was a complex of brick barracks built in late 1941 and early 1942 by prisoners of war. The structures were intended to be of a temporary character and were planned to be demolished after the end of the war. However, most of them have survived to this day. At present, there are 98 structures within the area of the former camp, including 76 masonry structures. The state of preservation of the structures concerned has been affected by a number of factors, including inter alia the quality of the materials used, the workmanship and the manner of utilisation. Bricks for the construction came from demolition of the surrounding villages, which explains their heterogeneity. The thickness of the walls is 12 cm, which promotes deterioration. The facilities did not have any horizontal waterproofing, which permitted capillary action up the walls from the beginning of their use. Furthermore, the facilities did not have any drainpipes that would protect the walls against the splashing of the water flowing down the roof and the exterior walls were not plastered. All of this meant that the walls of the buildings were and are still exposed to the action of water coming from precipitation and groundwater. During spring and autumn, the walls of these buildings alternately get saturated with water and dry out, which causes continuous migration of water within the texture of the material. In the long term and with prolonged presence of moisture, this process may result in degradation of the silicon-oxygen network of the vitreous phase. As a result, the texture of the brick undergoes detrimental changes [22] that cause reduction of its durability. In the winter time, when the temperature falls below 0 °C, water that is present in the pores of the materials freezes. This process is repeated cyclically in the period between autumn and spring over many years, even several times a day. During the daytime, water from the snow melting on the roofs flows freely down and splashes on the ground, causing heavy dampness of the wall, even up to half of its height. At night, when the temperature falls below 0 °C, water in the walls freezes. Figure 1 presents temperature changes in the region of the former extermination camp Auschwitz II-Birkenau illustrating the cyclical nature of this process. On this basis it can be estimated that the total number of freeze–thaw cycles for the bricks collected from those structures is at least in the range of several thousand. It is obvious that the more freeze–thaw cycles of water in the pores of the material, the more intensive the frost damage.

Based on macroscopic observations, three prevailing forms of damage of bricks in existing masonry camp buildings were distinguished; i.e., cracking, flaking and powdering. 

### 2.2. Purpose of the Research

In order to explain the causes of the various forms of damage, a research programme was carried out which included microscopic and petrographic observations as well as examination of the mineral composition and of the structure of porosity. Additionally, compressive and tensile strength testing was performed. The objective of the investigation was to determine the correlation between the structure of porosity, the mineral composition, the compressive strength and the form of damage of the bricks. Therefore, for each form of damage, three bricks were selected in different camp buildings as well as three bricks without any signs of damage. Due to the limitations stemming from the protection of the original substance of the heritage objects [23], the number and size of the samples collected for the testing were limited to the maximum. The places of collection of the samples were chosen in a manner enabling objective and representative evaluation of the materials. The samples were collected in the form of drill cores with a diameter of 50 mm (Figure 2). In total, 24 cores were collected and used to prepare appropriate samples for further laboratory testing. 

## 3. Methods

### 3.1. Photographic Documentation

Within the framework of the research investigation, extensive photographic documentation was performed with a NIKON D5100 camera (Tokyo, Japan), and this made the basis for the macroscopic description of the structures that are present within the area of the former extermination camp. Out of several hundred photographs, several were chosen which best reflected the observed forms of damage of the bricks.

### 3.2. Petrographic Observations by Optical Microscope

The purpose of the optical microscope observation was to characterise the mineral composition of bricks with different forms of damage. The analysis included identification of the components of the bricks and determination of their volume fractions. Attention was also paid to their size, habit and state of preservation. The examinations were carried out on 12 brick samples with defined forms of damage (including three undamaged ones). Standard preparation intended for polarized transmitted light microscopy was made from each sample. A thin slice was cut from each of the bricks and then embedded in resin with known refractive index and mounted on the microscope slide. Then, by grinding and polishing with diamond powder its thickness was reduced to 0.02 mm. Such prepared samples were examined using a petrographic optical microscope for polarized transmitted light microscopy JENAPOL (Carl Zeiss, Oberkochen, Germany) and the NIS-Elements BR 3.2 imaging software (Nikon Corporation). The volume fraction of components was determined in petrographic tests using the planimetric method (in line with a commonly applied procedure), by means of a point counter, by counting in each sample 600 points along four assumed measurement lines parallel to each other, distributed uniformly across the entire surface of the sample. The object stage travel distance, i.e., the distance between measurement points, was established individually for each sample depending on its grain size. The result of the analysis is the fraction of grain components and binder (identified using a microscope) expressed as a percentage of volume.

### 3.3. Observations by Scanning Microscopy with Elemental Analysis

The purpose of the observation of the microstructure was a qualitative identification of the crystalline and amorphous phases, their morphology and elemental composition. Like before, examinations were performed for 12 brick samples with defined forms of damage. The samples were in the form of irregular chunks with a volume of approx. 1 cm^3^. The specimens were glued onto stubs with silver glue. Due to the lack of sputtering (none coating), the examination was carried out in variable vacuum, in the pressure range of 80–120 Pa and at constant accelerating voltage EHT 20 kV. The microstructural observations were made in a scanning electron microscope EVO MA10 from Zeiss, equipped with VPSE, SE and BSD detectors and an X Flash 6/30 (EDS) detector from Bruker (Hamburg, Germany). 

### 3.4. Determination of the Structure of Porosity by Mercury Porosimetry

The purpose of the examination of the structure of porosity was to determine the porosity and the fraction of pores of a specific diameter. The examination was carried out on 12 samples with defined forms of damage. Testing was performed on samples with a volume of approx. 1 cm^3^ being in the air-dried state and dedusted. The examination of the structure of porosity was carried out using the method of mercury porosimetry in a Quantachrome Poremaster Nova 1000e porosimeter (Boynton Beach, FL, USA). The examination was carried out in the pressure range of 0.9–30 PSI with pore measurements in the range between approx. 10 nm and 300 µm.

### 3.5. Compressive Strength, Tensile Strength and Density Testing

In order to get the appropriate number of samples for compressive and tensile strength tests, two boreholes in the side face of the same brick were performed (directly in the wall). Samples with a diameter of 50 mm and a height of 50 mm were cut from cores collected in such a manner. (The other pieces of cores were used in petrographic and microstructural tests, etc.). In the samples intended for compressive strength testing, cylinder bases were ground in order to obtain smooth and parallel planes. Next, the samples were brought to the air-dry state. During the test, load was applied to the samples at 1 MPa/s. The tests were carried out in Zwick/Roell Z100 testing machine (Ulm, Germany).

In the case of samples intended for compressive strength testing (before their destruction), density in the air-dry state was determined.

The tensile strength test was also conducted using cylindrical samples: ø = h = 50 mm. At the mid height of the sample, the circumference of cylinders was notched to a depth of 5 mm in order to define tensile cross-section. Steel heads equipped with joints were stuck onto samples prepared in such a manner. In the tensile strength test, load was applied at 0.1 MPa/s. Tests were carried out in Zwick/Roell Z100 testing machine using samples in the air-dry state.

Compressive strength and tensile strength tests were carried out for each feature using 12 samples. In total, 24 samples representing the analysed forms of destruction were used in the tests. 

## 4. Results and Analysis

### 4.1. Macroscopic, Petrographic and Microstructural Characteristics of the Forms of Damage of Bricks

Based on the macroscopic observations, three basic forms of damage of ceramic bricks have been distinguished:Surface powdering progressing into the material; the product of disintegration of the material is brick crumbled into powder. The samples were labelled as P1, P2 and P3;Surface flaking progressing into the material; distinct layers of flakes occur on the surface and fall off. The samples were labelled as F4, F5 and F6;Cracking, which takes place within the volume of the brick and is not only associated with the superficial layers. This results in formation of deep scratches and consequently chunks of material that wedge each other and fall off after some time. The samples were labelled as C7, C8 and C9.

For each of the described forms of damage of bricks, a macroscopic, petrographic and microstructural image was presented (Figure 3, Figure 4 and Figure 5). For comparison, corresponding observations were made for bricks without signs of damage (Figure 6) which were labelled as S10, S11 and S12. Table 1 presents the mineral composition of the tested bricks. 

Brick powdering leads to the formation of fine ceramic powder that is systematically removed from the surface of the material by its own action. The process of deterioration begins on the surface and gradually advances inwards. In the camp structures that were studied, this sort of damage is relatively commonly observed. Microstructural studies demonstrated that the characteristic feature of this sort of damage is a loosened, fine-crystalline texture. The structure of the ceramic mass is homogeneous, directional and microporous. Its directionality is expressed by the parallel arrangement of the elongated mineral grains and rock crumbles. The pores have elongated and irregular shapes. Occasionally, they are formed as thin and short crevices. Also present are irregular crevices with a length reaching 1.5 mm and opening up to 0.1 mm. The ceramic mass (binder) is in a large part composed of thermally modified clay substance with cryptocrystalline internal structure and of iron hydroxides. It is fairly evenly distributed between the grains forming thin compact envelopes around them. Grain material is mainly represented by mono- and polycrystalline quartz. Crumbles of quartz-sericite schist are sparse and always have large sizes reaching 0.6 mm. Typically, they have rectangular and oval shapes, and are rather well preserved. Plagioclases form small grains with rectangular or irregular shapes and sizes not exceeding 0.15 mm. They are usually corroded. The grains do not contact each other or only do so pointwise. Sericite is quite common as grain component. Plagioclases form small grains with rectangular or irregular shapes and sizes not exceeding 0.15 mm. They are usually corroded. The grains do not contact each other or only do so pointwise. Sericite is quite common as grain component. Its plates are thin and long, and quite fine. A large proportion of such components of low mechanical durability, which are additionally characterised by low cohesion to other mineral components, may cause crumbling and powdering of small fragments of bricks. On the other hand, concentrations of fine-crystalline calcium carbonate are few. They have oval shapes and small sizes, below 0.1 mm. They are distributed disorderedly and do not exhibit any zonation. 

The second form of damage, also advancing from the surface, is flaking. In this case, clear layers of flakes form on the surface, and over time they delaminate and fall off. In the microstructural examinations, stacked layers of material with sizes in the range of several dozen micrometres were observed. The texture of this kind of bricks is distinctly compact and more consolidated. The structure of the ceramic mass is directional and microporous. It is expressed in the parallel arrangements of the elongated mineral components and pores. The prevailing pores are of elongated or irregular shapes. The basic mass of the brick (binder) consists of clay minerals unevenly distributed between the grains. They form dark brown smudges of binder of varying thickness and length. Between them there are present irregular, dark brown concentrations and smudges of clay substance with elevated fraction of iron compounds. Iron hydroxides also form separate concentrations of sizes reaching 0.15 mm. Concentrations of clay minerals are arranged in a flat-parallel manner together with other mineral components, accentuating the directional structure of the brick. In accordance with this direction, crevices are formed, present particularly in the external part of the samples. The grain material is composed of mono- and polycrystalline quartz. Crumbles of quartz–sericite schist are sparse and have small sizes, not exceeding 0.15 mm. Plagioclases and alkali feldspars are sparse. They form irregular and sharp-edged grains of small sizes (up to 0.1 mm), typically substantially altered chemically and corroded. Mineral grains do not contact each other or show rare pointwise contacts. Sericite is present in the form of usually fine plates although also larger forms can be found, with length reaching 0.25 mm. A large quantity of this component is the cause that the ceramic mass may exhibit directional divisibility and disintegrate along the smooth surfaces of sericite blades. This kind of phenomenon is observed in the boundary sections of brick samples with this form of damage.

The last distinguished form of damage, i.e., cracking, is related to the forming of sharp chunks sized from a few to over a dozen mm. This process takes place not only on the surface but also in the deeper layers. Initially, the formed chunks are tightly wedged in the material; however, over time they gradually come loose as a result of, e.g., water freezing in the crevices formed or microbial growth. This leads to the expansion of the crevices that had formed, which causes chunks of bricks to fall off. The ceramic mass has a directional and microporous structure. Within it there are alternately arranged irregular, smudged layers composed of clay material with a different composition that has light and dark brown colour in microscope images. The directionality of the structure is also expressed by the presence of siliceous veins with thickness below 0.1 mm, occasionally pinching out and vanishing. At the boundary between the layers of different mineral composition or their boundary with the siliceous veins there may occur cracking of the brick, which is evidenced by the elongated crevices that form in such places. The aforementioned structure is also accentuated by thin sericite blades arranged in a flat-parallel manner as well as elongated grains of quartz. The prevailing pores are of irregular and elongated shapes. The diameter sizes are varied, typically not exceeding 0.1 mm and only sporadically reaching 0.2 mm. The basic mass of the brick is made of thermally treated clay substance that is composed of thin layers or irregular concentrations of a light brown colour arranged alternately with layers of dark brown colour. The grain material is mainly quartz in the form of monocrystalline, and more rarely polycrystalline grains. Quite numerous are bundles of blades of sericite formed from degradation of micas. They have small lengths reaching 0.15 mm and are usually slightly delaminated without any clear indications of transformation. Plagioclases and alkali feldspars, represented mainly by orthoclase, are present in a small quantity. These are small grains with indications of slight changes and corrosion. The grains do not contact each other and are enclosed by components of the binder. Apart from grain components, presence of small oval concentrations of iron hydroxides was found, as well as some charred organic matter, which was not dispersed in the course of homogenisation of the ceramic material. Present in the brick are also concentrations of fine-crystalline calcium carbonate originating most probably from the mortar. They have sizes reaching a maximum of 0.35 mm as well as oval and round shapes. 

Bricks without any signs of damage are characterised by a microporous, isotropic structure. Present are pores with diameters not exceeding 0.1 mm (they only sporadically reach 0.3 mm) and irregular shapes. The ceramic mass constituting the binder is composed of thermally modified clay substance with cryptocrystalline internal structure, tinted with iron oxide pigment. The aggregate is chiefly made of mono- and polycrystalline quartz. A large fraction of binder means that the grains hardly contact each other. Plagioclases and alkali feldspars, represented by orthoclase, typically form grains of small sizes but large specimens of these minerals, reaching 0.25 mm, are also present. They are poorly enveloped and quite clearly corroded. Sericite is sparse and arranged disorderly. It forms fine blades, substantially altered chemically, with faded bundle structure. There are no contacts between the mineral grains or occasional pointwise contacts are present. Crumbles of clay with sizes reaching 0.5 mm are oval and well enveloped. Based on observations in the microstructural examinations, presence of the vitreous phase was discovered. 

### 4.2. The Structure of Porosity

The results of the examination of the structure of porosity by means of mercury porosimetry have been presented in the form of accumulation and population curves (Figure 7, Figure 8, Figure 9 and Figure 10). 

Due to the lack of a clear-cut division of pores according to their diameters with regard to their impact on frost resistance, the authors developed their own categorisation enabling evaluation of the correlation between pore diameters and the forms of frost damage. Based on the population curves, five categories of pores were distinguished and their respective content in all the bricks tested was determined. On this basis, the fraction of pores prevalent in bricks characterised by the specific forms of frost damage were determined (Figure 11).

Based on the pore volume distributions obtained, it was demonstrated that bricks without any signs of damage and powdering bricks are clearly characterised by the prevalence of pores with diameters in the range of 1–10 µm. On the other hand, in bricks with signs of flaking and cracking the prevailing category of pores is that in the ranges of 0.1–1.0 µm and 1–10 µm, while the fractions of the individual categories depending on the form of damage is varied. 

The test results concerning the structure of porosity showed lack of frost resistance for bricks in which the predominant pores have diameters below 1 µm, which is consistent with the studies [14]. On the other hand, for bricks without any signs of damage, the predominant pores are in the range of 1–10 µm the fraction of which is approx. 90%. Bricks with the highest strength value, i.e., 40 MPa, have over 90% of pores with diameters in the range of 3–10 µm. According to [1,13,14], such a structure of porosity ensures frost resistance. 

Samples of bricks characterised by damage in the forms of cracking and flaking display similarities with regard to their structures of porosity. In both cases, there are pores with diameters below 0.1 µm present. Their fraction is significant. In the case of damage by cracking it is between 10% and 22% while for the damage form of flaking it is between 4% and 11%. For both forms of damage, the characteristic feature is the predominant fraction of pores with diameters below 1 µm, which is between 56 and 85% of the entire pore population for the cracking damage and between 27 and 40% for the flaking damage. These results confirm low frost resistance of bricks with a significant proportion of pores with small diameters [13,14]. 

The structure of porosity of the bricks that display damage in the form of powdering is comparable to the structure of porosity of the bricks without signs of damage. The similarity consists of the lack of the smallest pores, i.e., with diameters below 0.1 µm, and the prevailing proportion of pores in the range 1–10 µm. However, in the powdering bricks, more pores with diameters over 10 µm were found, as well as more pores with diameters below 1 µm than in the bricks that had no signs of destruction. 

### 4.3. Results of Compressive Strength, Tensile Strength and Density Testing 

The results of compressive strength, tensile strength and density testing are shown in Table 2.

According to ASTM [24], depending on the degree of aggressiveness of the environment, brickwork should be characterised by a minimum compressive strength of 211 kg/cm^2^ (20.7 MPa) for bricks resistant to severe weathering, 176 kg/cm^2^ (17.3 MPa) for bricks resistant to moderate weathering and 106 kg/cm^2^ (10.4 MPa) for bricks resistant to negligible weathering. The results obtained unequivocally demonstrated that an appropriately high strength of bricks (acc. to ASTM) does not guarantee frost resistance. Furthermore, compressive strength tests demonstrated that bricks characterised by powdering have the lowest strength values. This is characteristic for this form of damage. With regard to the other bricks, the results of compressive strength tests are similar and clearly higher. It is worth noting that the strength of the bricks without signs of frost damage is similar to that of the bricks that were damaged in the form of flaking and cracking. 

## 5. Discussion of Results

Frost resistance of bricks depends on many factors and above all on the structure of porosity, the mineral composition and the strength [1,2,6,9,15]. Like frost resistance, the form of frost damage of bricks is also the result of several factors. Figure 12 presents the mutual correlations between the material characteristics that have impact on the form of frost damage of the bricks tested. A form of frost damage (FFD) factor was introduced as a parameter describing the structure of porosity, being the ratio of the percentage of pores with diameters of 3–10 µm to the total content of pores with diameters of 0–10 µm (Equation (1)). According to [1,2,7], the total content of pores with diameters of 0–10 µm, taking into account the content of pores with diameters of 1–3 µm and 3–10 µm, seems to best reflect the impact of the structure of porosity on the form of frost damage of bricks.
(1)$$FFD=\;\frac{\sum_{volume\;of\;pores}^{}3-10\;µm}{\sum_{volume\;of\;pores}^{}0-10\;µm}$$

In order to find the factor that has the greatest impact on the form of frost damage (FFD factor), an attempt was made to establish quantitative correlations between the strength, the structure of porosity and the mineral composition (binder content) for forms of damaged that had been distinguished. Figure 13 presents the mutual correlations between these characteristics. 

The bricks labelled as S, which do not display signs of frost damage, are characterised compressive strength at the level of 20–45 MPa, a high content of binder (over 50%) and a very high percentage of pores in the range of 0–10 µm (close to 90%) as well as a very high FD factor at the level of 75–90. A characteristic feature of these bricks is an isotropic and homogeneous structure, which ensures the isotropy of the mechanical properties. The co-occurrence of the above qualities guarantees frost resistance of a brick over a very long period of time. 

When analysing the results obtained, it can be observed that the least variable parameter in all the bricks was the binder content. In the samples tested it is within the range of 40–55% where in the damaged bricks the content of binder does not exceed 50% while in the undamaged ones it is greater than 50%. The lack of appropriate volume of matrix can be the reason for the weakening of the texture of the material, poor grain embedment and bad envelopment, which consequently makes the material more susceptible to frost damage. Thus, a drop in the binder content affects frost resistance; on the other hand, no effect on the form of frost damage has been observed. The content of binder in the damaged bricks, regardless of the form of frost destruction, is similar. 

Another parameter characterising the bricks tested is their compressive strength. For the samples without signs of frost damage S as well as for the samples characterised by damage in the forms of flaking F and cracking C, the compressive strength is similar and in the range between 20 to 45 MPa. A direct correlation between the strength and the form of frost damage has not been observed. An exception is the powdering bricks P, which are characterised by significantly lower strength values, in the range of 10–15 MPa. It can thus be concluded that the occurrence of this form of frost damage is conditional on low strength of bricks, not exceeding 20 MPa. 

Therefore, it seems that the essential and most important parameter impacting the form of damage is the structure of porosity. However, a clear correlation has been observed between the form of damage and the content of pores of diameter 3–10 µm. The parameter that best describes the form of frost damage relative to the structure of porosity is the FFD factor (Figure 10) which falls within the range of 75–90 for the samples without signs of frost damage S. For the flaking samples (F) the FFD factor is 4–10, for the cracking (C) ones the FFD factor is 2–11, and for the powdering samples (P) the FFD factor is 45–50. Similar value of FFD factor for cracking and flaking samples with the same binder contents and similar compressive strength, show the reasons for the varied forms of damage should be sought in the spatial arrangement of the texture components. Optical microscope observations have shown that the samples that were affected by cracking were also characterised by a layered structure of the ceramic mass. At the boundary between the layers of different mineral composition or at their boundary with the siliceous veins, crevices occur. Then, more of less regular fragments of the brick may get loosened along them. Also, flaking is related to the directional structure of the ceramic mass as expressed by the flat-parallel arrangement of mineral grains. In the case of this mechanism of damage, as well as in the case of powdering, the process of damage seems to be intensified by the presence of sericite. The scaly habit of the sericite grains and smooth and slippery surfaces of the walls promote flaking. Along the planes determined by the accumulation of such plates arranged in a flat-parallel manner to each other there may occur disintegration of the brick under load.

## 6. Conclusions

Both frost resistance and the form of frost damage of bricks is the result of several factors, such as the mineral composition, the structure of porosity and mechanical strength. The investigation presented in the paper enabled pointing out the material properties that have a crucial impact on the form of frost damage. It has been shown that bricks get damaged in a specific way depending on their structure of porosity, mineral composition and spatial arrangement of the texture components. Among these, it is the structure of porosity that has the dominant influence. 

The tests have not revealed a correlation between the strength of the bricks and their frost resistance or the forms of damage. Both the damaged and the undamaged bricks were characterised by strength values at a similar level, i.e., in the range between 20 and 40 MPa. An exception was the powdering bricks which had significantly lower strength than the other bricks. This suggests that bricks with low strength are susceptible to this form of frost damage. A characteristic common to all the bricks that were affected by frost damage is a lower content of binder combined with elevated content of sericite as compared with the undamaged samples. On the other hand, no correlations have been found between the mineral composition of the bricks tested and their forms of frost damage.

The factor that essentially differentiates the characteristics of the bricks tested is the structure of porosity. It has been demonstrated that bricks with a relatively high share of pores with diameters smaller than 1 µm in the total population of pores undergo frost damage; i.e., they are characterised by a lack of frost resistance. Under the influence of cyclical freezing and thawing in actual conditions, these bricks undergo damage but the form of the damage, i.e., cracking, flaking or powdering, depends above all on the structure of porosity, i.e., the fraction of pores of specific diameters. It has been observed that the form of frost damage is highly correlated with two categories of pores, i.e., pores with diameters of 1–3 µm and 3–10 µm. On this basis, the form of damage factor, FFD, was introduced as the ratio of the percentage of pores with diameters of 3–10 µm to the total content of pores with diameters of 0–10 µm. It is highest for samples without signs of damage S and amounts to between 75 and 90; for the powdering P form of frost damage it is 45–50. For the flaking F forms, it is in the range 4–10 and for the cracking C forms it is the lowest, in the range of 2–11. 

Based on the results obtained, it has been demonstrated that parameters such as the structure of porosity and mineral composition of bricks may constitute a basis for assessing frost resistance of bricks. In structures of special historical value for which protection of the original substance is a priority and collection of large-sized samples is very problematic, it is necessary to consider predicting durability of the materials based on structural studies. Their invaluable advantage is that they can be performed on small fragments of the material, which essentially reduces tampering with the protected heritage substance.

## Figures and Tables

**Figure 1 materials-12-01165-f001:**
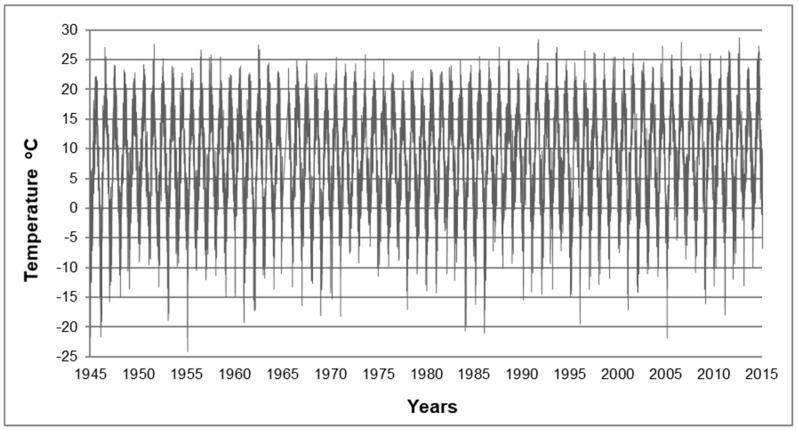
The course of average daily temperature values recorded at a meteorological station located a few kilometers from Auschwitz. Data Institute of Metrology in Warsaw.

**Figure 2 materials-12-01165-f002:**
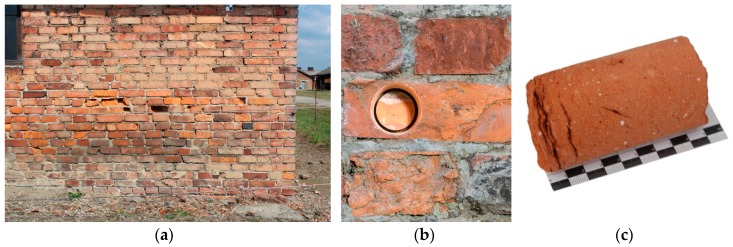
(**a**) Part of a wall of a prisoners’ barracks, (**b**) the place of sampling, (**c**) the collected brick core.

**Figure 3 materials-12-01165-f003:**
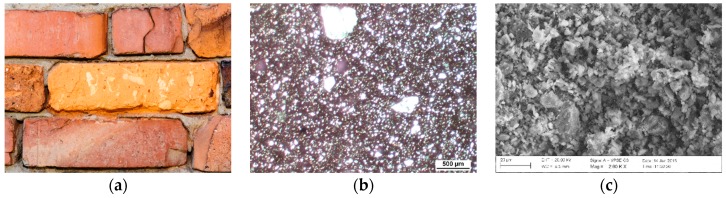
Macroscopic view of damage of the surface of the bricks as a result of surface powdering (**a**). Transmitted light optical microscopy image. Homogeneous structure of the ceramic mass (**b**). Microstructure of bricks damaged as a result of surface powdering (**c**).

**Figure 4 materials-12-01165-f004:**
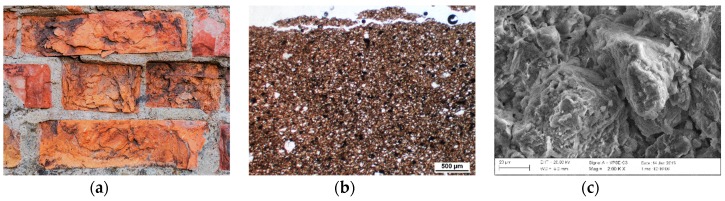
Macroscopic view of damage of the surface of the bricks as a result of surface flaking on the inside of the sprinkler (**a**). Optical microscope image in transmitted light. Crevice pattern matching the arrangement of the ceramic mass layers (**b**). Microstructure of bricks damaged as a result of surface flaking (**c**).

**Figure 5 materials-12-01165-f005:**
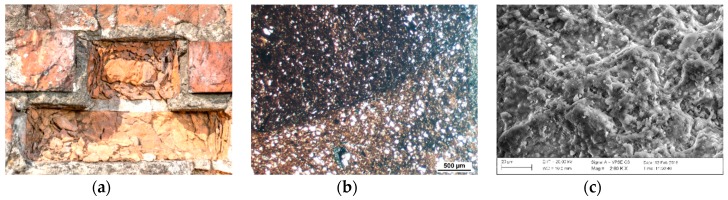
Macroscopic view of damage of the surface of the bricks as a result of cracking (**a**). Optical microscope image. Directional structure of the ceramic mass. Visible are smudges and thin layers containing binding agent with varied fraction of clay substance and iron compounds. Slight directionality of the ceramic mass and a crevice with a pattern matching the component arrangement (**b**). Microstructure of bricks damaged as a result of surface cracking (**c**).

**Figure 6 materials-12-01165-f006:**
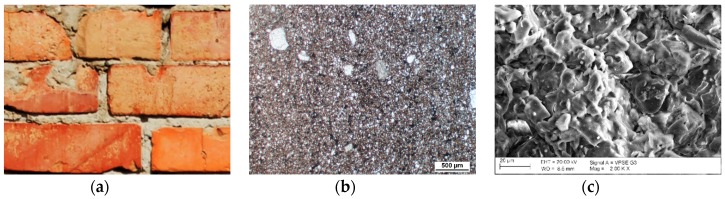
Macroscopic view of a brick without signs of damage (**a**). Optical microscope image of a brick with compressive strength of 40 MPa. Abundant binder between the grain components (**b**). Microstructure of bricks without signs of damage, visible phases with clear vitrification (**c**).

**Figure 7 materials-12-01165-f007:**
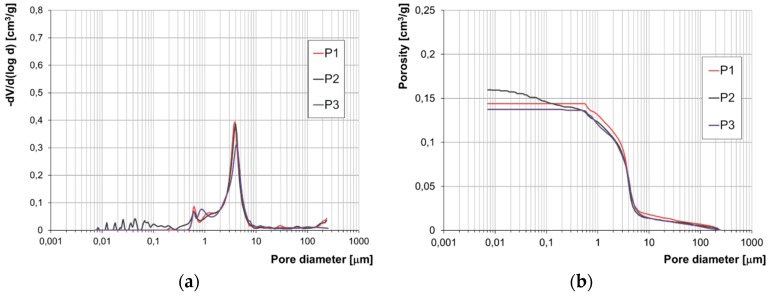
Pore size distribution (**a**) and cumulative curves of pores volume distribution (**b**) for powdering bricks P1, P2 and P3.

**Figure 8 materials-12-01165-f008:**
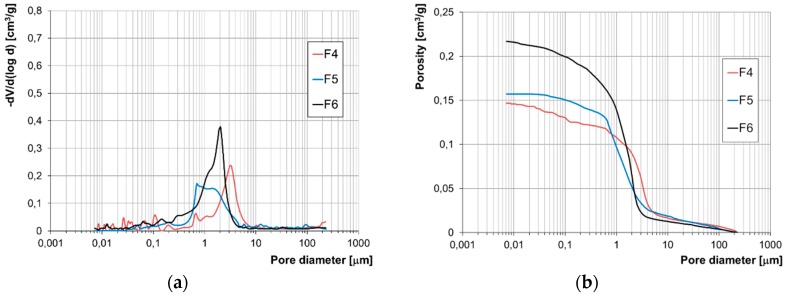
Pore size distribution (**a**) and cumulative curves of pores volume distribution (**b**) for flaking bricks F4, F5 and F6.

**Figure 9 materials-12-01165-f009:**
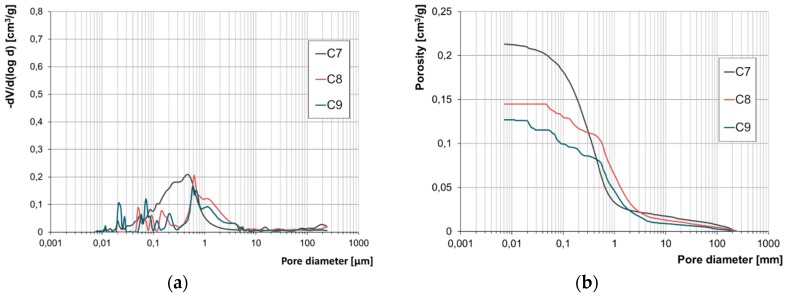
Pore size distribution (**a**) and cumulative curves of pores volume distribution (**b**) for cracking bricks C7, C8 and C9.

**Figure 10 materials-12-01165-f010:**
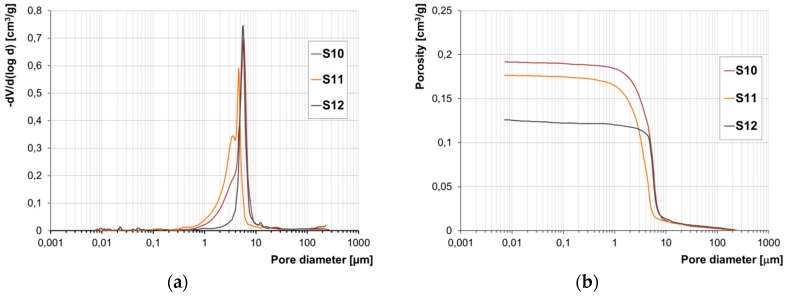
Pore size distribution (**a**) and cumulative curves of pores volume distribution (**b**) for bricks without signs of damage S10, S11 and S 12.

**Figure 11 materials-12-01165-f011:**
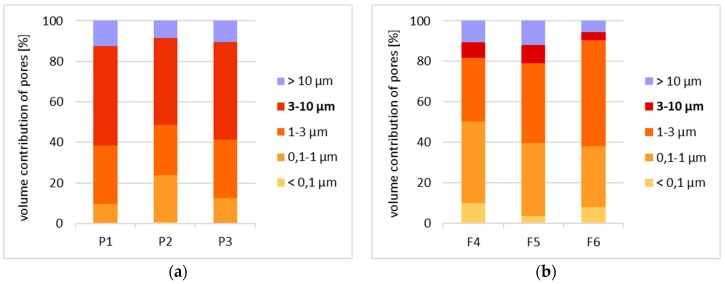
Classification of pores in bricks damaged in the form of powdering P1, P2 and P3 (**a**), flaking F4, F5 and F6 (**b**), cracking C7, C8 and C9 (**c**) and standard S10, S11 and S12 (**d**).

**Figure 12 materials-12-01165-f012:**
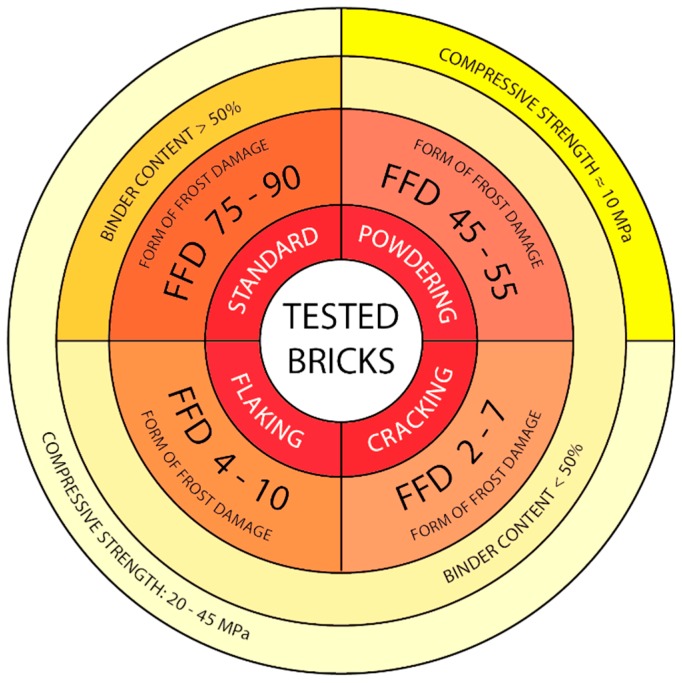
Material characteristics having an impact on the form of frost damage. The closer to the centre of the graph, the higher the force of impact of a given factor on the form of damage.

**Figure 13 materials-12-01165-f013:**
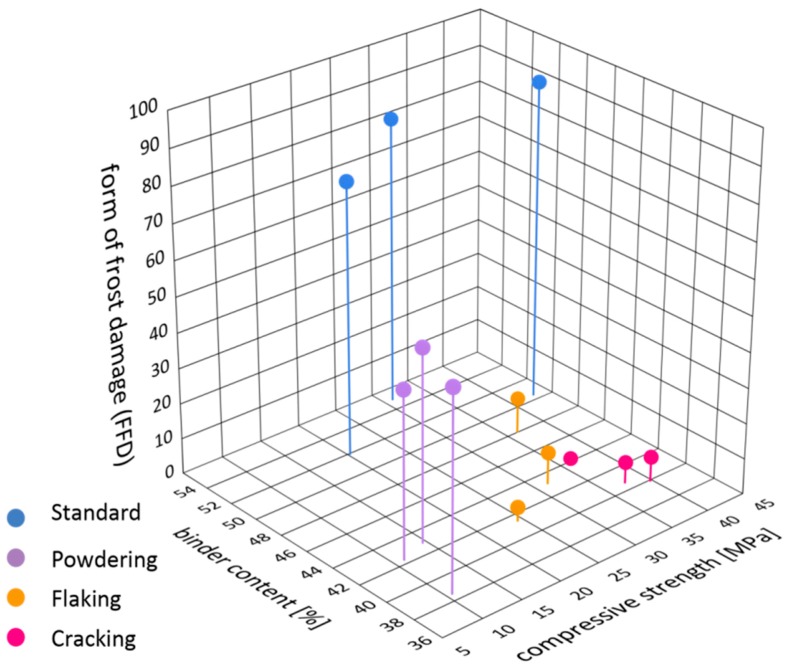
Quantitative correlation between compressive strength, binder content and form of frost damage (FFD) factor.

**Table 1 materials-12-01165-t001:** Fractions of mineral components in bricks with various forms of damage.

Component	Mineral	Powdering			Flaking			Cracking			Standard		
		P1	P2	P3	F4	F5	F6	C7	C8	C9	S10	S11	S12
		Volume Fraction [%]											
Mineral grains	Quartz	31.2	31.3	31.3	35.2	23.4	34.5	26.3	24.3	31.0	28.9	32.4	31.9
	Alkali feldspars	2.9	0.0	0.0	0.7	0.0	0.0	0.3	0.0	4.2	1.9	2.0	0.0
	Plagioclase	1.5	3.0	0.9	1.4	0.0	4.6	2.4	0.6	0.0	3.9	0.0	4.7
	Sericite	22.9	16.9	6.6	17.1	16.1	14.4	15.3	29.1	7.4	5.8	5.6	5.9
Crumbs of rock	Chert	0.0	0.0	1.0	0.0	3.7	0.0	0.0	0.0	1.4	0.0	0.0	0.0
	Slate quartz-sericite	0.0	1.5	0.0	0.7	0.0	0.0	0.0	0.0	0.0	0.0	0.0	0.0
	Clay	2.0	0.0	8.1	0.0	5.1	5.2	0.0	0.0	6.1	4.8	5.0	7.0
Aggregates	-	60.5	52.7	47.9	60.1	48.3	58.7	37.4	54.0	50.1	45.3	45.0	50.5
Aggregates of the other components	Hydroxides of iron	1.0	5.0	5.0	1.8	1.5	0.0	2.1	3.3	1.9	0.0	0.0	0.0
	Charred organic substance	0.0	0.0	0.0	0.0	2.2	0.0	3.2	1.2	2.8	2.9	0.0	0.0
	Calcite	0.0	0.0	4.4	0.0	0.0	0.0	6.7	0.9	5.6	0.0	0.0	0.0
Binder	-	38.5	42.3	42.9	43.1	48.0	41.4	43.7	40.6	39.9	51.9	54.9	50.4
Sum	-	100	100	100	100	100	100	100	100	100	100	100	100

**Table 2 materials-12-01165-t002:** Results of compressive strength, tensile strength and bulk density testing.

Form of Damage	Sample Identification	Compressive Strength [MPa]	Tensile Strength [MPa]	Bulk Density [g/cm^3^]
Powdering	P1	10.6	1.3	1.69
	P2	11.1	0.9	1.63
	P3	14.5	1.5	1.67
Flaking	F4	31.5	3.6	1.80
	F5	36.3	4.1	1.78
	F6	24.3	1.9	1.67
Cracking	C7	35.8	3.2	1.79
	C8	37.5	3.8	1.69
	C9	39.7	3.3	1.81
Standard	S10	20.2	1.7	1.76
	S11	31.1	3.8	1.76
	S12	43.0	5.1	1.86

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
