# Peer review of "Forms of Damage of Bricks Subjected to Cyclic Freezing and Thawing in Actual Conditions"

_materials, 2019, doi:10.3390/ma12071165_

Round 1
Reviewer 1 Report
I found this work very interesting and inspiring.
The objective and scope is “hidden” in the text. I will recommend adding research questions to the text.
Section 3.5: I find this section unclear. Did you test compressive and tensile strength on undamaged or damaged brick? I understand from the text that you tested 9+9=18 cores, but the total number of core samples are 12. Did you test compressive and tensile strength on the same cores? I miss a clearer description of the tensile test setup.
Section 4.2: You present information about the porosity distribution and the total porosity. I find it of outmost interesting for the objective of the paper to know the water absorption (WA) of the brick.
Conclusions (line 431-433): Are you sure the brick was not weakened by the frost degradation?
Minor corrections:
Table 1: Are “Sum”’ = “Aggregate” + “Aggregates of the other components” + “Binder”? The sum is not 100.
Figure 7 – 10 a): What is the y-axe? Please explain with words
Line 10 to 15: Too long sentence. Please split.
Line 303: This is section 4.2
Line 348: This is section 4.3
Line 364: Header “Discussion”. Drop “of results”
Author Response
Dear Reviewer,
Please find attached a document containing our responses.

Reviewer 2 Report
The article is interesting, but there are uncertainty moments:
Disintegration of clay bricks may have occurred, not only because of frost resistance. Working loads, wind, ultraviolet rays, acid rain, which can cause various chemical and physical processes in the materials, are also important. It would be useful to determine the frost resistance of such bricks in the laboratory.
It is not logical that undamaged bricks have less strength than damaged bricks. It must be provided more detailed description of the strength determination and photos of specimens. What's more, the amount of binder in undamaged bricks is the highest. Maybe there was a different technology for brick manufacture. These questions have to be analysed.
Line 22?
How was determined volume fractions of minerals?
Fig. 8-9. F1-F3? Titles of Figures are not correct.
In Fig. 12 almost all information is repeated.
Author Response

(The authors gave the same response as above.)

Round 2
Reviewer 2 Report
In the paper have to be explanation about compressive strength, particularly why not damaged bricks have less compressive strength, than damaged.
Author Response
Dear Reviewer,
Thank you for your suggestions. Below there is an answer for your comments. We also considerably improved the description of compressive and tensile strength tests (section 3.5).
Kind regards
Teresa Stryszewska
In the paper have to be explanation about compressive strength, particularly why not damaged bricks have less compressive strength, than damaged.
The obtained strength (using the method described by us) was the strength of the ceramic material in the brick. For that reason, the obtained results were comparative in nature and enabled an assessment of a given population of results of strength obtained by means of the same method. There were a lot of not destructed bricks in the buildings that we tested, the strength of which was in the range of 20–40 MPa, sometimes more. For that reason, we chose for the article bricks representing an entire population of bricks, i.e. 20, 30 and 40 MPa, in order to determine the impact of strength or lack thereof on the form of destruction. Nevertheless, observations and tests showed that some of the bricks with relatively high strength underwent destruction, whereas some were not destructed. This clearly points to the fact that strength is not closely related to frost resistance or the form of destruction (which was showed graphically in Fig. 12). Frost resistance is the result of several factors: porosity structure, binder content, composition and strength. However, it depends primarily on the structure of porosity which has the biggest force of impact on the form of frost destruction.